# Construction of *Candida albicans* Adhesin-Exposed Synthetic Cells for Preventing Systemic Fungal Infection

**DOI:** 10.3390/vaccines11101521

**Published:** 2023-09-25

**Authors:** Zirun Zhao, Ying Sun, Mingchun Li, Qilin Yu

**Affiliations:** 1Key Laboratory of Molecular Microbiology and Technology, Ministry of Education, Department of Microbiology, College of Life Sciences, Nankai University, Tianjin 300071, China; 2120221390@mail.nankai.edu.cn (Z.Z.); 2120231588@mail.nankai.edu.cn (Y.S.); nklimingchun@163.com (M.L.); 2Research Center for Infectious Diseases, Nankai University, Tianjin 300350, China; 3National Key Laboratory of Intelligent Tracking and Forecasting for Infectious Diseases, Tianjin 300350, China

**Keywords:** fungal vaccine, systemic fungal infection, *Candida albicans*, adhesin, synthetic biology, protection

## Abstract

The development of efficient fungal vaccines is urgent for preventing life-threatening systemic fungal infections. In this study, we prepared a synthetic, cell-based fungal vaccine for preventing systemic fungal infections using synthetic biology techniques. The synthetic cell EmEAP1 was constructed by transforming the *Escherichia coli* chassis using a de novo synthetic fragment encoding the protein mChEap1 that was composed of the *E. coli* OmpA peptide, the fluorescence protein mCherry, the *Candida albicans* adhesin Eap1, and the C-terminally transmembrane region. The EmEAP1 cells highly exposed the mChEap1 on the cell surface under IPTG induction. The fungal vaccine was then prepared by mixing the EmEAP1 cells with aluminum hydroxide gel and CpG. Fluorescence quantification revealed that the fungal vaccine was stable even after 112 days of storage. After immunization in mice, the vaccine resided in the lymph nodes, inducing the recruitment of CD11c^+^ dendritic cells. Moreover, the vaccine strongly activated the CD4^+^ T splenocytes and elicited high levels of anti-Eap1 IgG. By the prime-boost immunization, the vaccine prolonged the survival time of the mice infected by the *C. albicans* cells and attenuated fungal colonization together with inflammation in the kidneys. This study sheds light on the development of synthetic biology-based fungal vaccines for the prevention of life-threatening fungal infections.

## 1. Introduction

With an increased number of immunocompromised individuals suffering from viral infections, radiotherapy/chemotherapy, antibiotic abuse, and organ transplantation, human beings are at high risk for life-threatening fungal infections [1,2,3]. In clinics, fungal pathogens (e.g., *Candida albicans*, *Candida auris*, *Aspergillus fumigatus*, *Cryptococcus neoformans*, etc.) are becoming prevalent. *Candida*-related infections in particular, which are frequently associated with candidemia and multi-organ damage, contribute to about 40% of hospital-acquired fungal infections with high mortality [4,5,6,7]. Up to now, unfortunately, there are only a few kinds of antifungal drugs such as azoles, amphotericin, and echinocandins [8]. The appearance of more and more drug-resistant *Candida* pathogenic strains is impacting current antifungal therapies. These strains take the lives of millions of people every year [9,10]. Therefore, it is urgent to develop novel strategies against fungal infections.

Vaccination is no doubt a promising way to protect individuals from fungal infections [11,12]. Nowadays, there are a few kinds of preventive fungal vaccines that have been developed [13,14,15,16]. For example, live-attenuated or killed fungal cells of *C. albicans*, *Saccharomyces cerevisiae*, and *C. neoformans* were found to induce bodily production of antibodies against real fungal pathogens [17,18,19]. The recombinant protein fragments of *C. albicans* Sap2, Als3, Hsp90, or Hyr1 could induce production of specific antibodies to neutralize pathogen surface proteins [20,21]. Fungal cell wall polysaccharides, or their mimics (e.g., mannans, β-glucans, laminarins, etc.), could also be used as antigens to induce protective immunity against fungal infections [20]. However, most of these fungal vaccines have failed to be widely accepted by customers, owing to their high costs, low protective efficiency, or strong side effects. Nowadays, it remains a great challenge to easily prepare economic, efficient, and safe fungal vaccines.

The development of synthetic biology brings a great opportunity for the preparation of novel vaccines [22]. By artificial design of protein antigen-encoding modules, polysaccharide antigen-processing circuits, or even safe virus genomes, a series of antigens, DNAs, and mRNAs could be obtained by fermentation engineering on a large scale [23,24,25]. These products are economically used as raw materials for different vaccines for the protection of the body from pathogen infections and tumorigenesis [26,27]. Among them, the SARS-CoV-2 mRNA vaccine facilitated by the synthetic biology strategy is becoming a great weapon of human beings for fighting against the virus [28,29]. Unfortunately, there are no promising fungal vaccines based on synthetic biology techniques for application in the prevention of fungal infections.

*C. albicans* Eap1 is a critical adhesin that is highly expressed in both yeast and hyphal cells and is efficiently exposed on the surface of fungal cells [30,31]. In this study, we aimed to develop one kind of fungal vaccine through the construction of *C. albicans* adhesin Eap1-exposed synthetic cells. A fusion protein composed of the mCherry fluorescence protein and the Eap1 fragment named mChEap1 was designed and exposed on *E. coli* chassis cells with the aid of the secretion peptide OmpA. The constructed synthetic cells were then mixed with the immune activator CpG and aluminum hydroxide gel, obtaining the final fungal vaccine AlgelCpG + EmEAP1.

## 2. Materials and Methods

### 2.1. Materials

The aluminum hydroxide gel was purchased from Aladdin (Aladdin Biochemical Technology Co., Ltd., Shanghai, China). The TLR9 agonist CpG 1826 (5′-TCCATGACGTTCCTGACGTT-3′) was synthesized by Sangon Biotechnology (Sangon Biotech Co., Ltd., Shanghai, China). The *E. coli* strain BL21 (the chassis) was obtained from GenScript (GenScript Biotech Corp., Nanjing, China). To prepare the anti-Eap1 antibody, rabbits were immunized with the purified Eap1 protein from genetically engineered *E. coli* cells via the prime-boost procedure, followed by serum isolation and antibody purification. The obtained antibody was further labeled by FITC in 10 mM NaHCO_3_ solution (pH = 9.0). DAPI and FITC were purchased from Sigma (Sigma Technology Co., Ltd., Sterling Heights, MI, USA). The Cy5 NHS ester was purchased from APExBIO (APExBIO Technology LLC., Houston, TX, USA). The FITC-conjugated anti-CD11c and anti-CD4 antibodies, together with the APC-conjugated anti-mouse IL-2 and IFN-γ antibodies, were purchased from BioLegend (BioLegend Biotechnology Co., Ltd., San Diego, CA, USA). The HRP-conjugated anti-mouse IgG antibody was purchased from Santa Cruz (Santa Cruz Biotechnology Co., Ltd., Paso Robles, CA, USA). The ELISA colorization agents were obtained from Dingguo (Dingguo Biotechnology Co., Ltd., Beijing, China). The mouse TNF-α assay kits were purchased from Jianglaibio (Jianglai Biotechnology Co., Ltd., Shanghai, China).

### 2.2. Construction and Observation of the Synthetic Cells

To construct the synthetic strain EmEAP1, the artificial gene *OmpA-mCherry-EAP1-CTR* was first designed using the optimized codons of *E. coli*, based on the sequence of the corresponding encoded protein mChEap1. The designed gene was then de novo synthesized by GenScript (China) and then cloned into the plasmid pET-28a, obtaining the plasmid pET-*OmpA-mCherry-EAP1-CTR*, in which the *OmpA-mCherry-EAP1-CTR* gene was under the control of the strong T7 promoter and the T7 terminator. The plasmid was then transformed into the BL21 strain, obtaining the synthetic *E. coli* strain EmEAP1. As the control strain, Em was constructed by transformation of the BL21 strain with the plasmid pET-*OmpA-mCherry-CTR* in which the *EAP1* sequence was absent. Both Em and EmEAP1 cells were cultured in the liquid LB medium containing 0.5 mM IPTG for 12 h, followed by staining with DAPI (10 mg/L) and the FITC-conjugated anti-Eap1 antibody (20 mg/L) for 20 min. The stained cells were then observed by confocal microscopy (FV1000, Olympus Corp., Tokyo, Japan). The fluorescence intensity units (FLU) of FITC of the stained cells was determined by using a fluorescence microplate reader (Enspire, PerkinElmer Inc., Waltham, MA, USA). 

### 2.3. Preparation and Observation of the Fungal Vaccines

The fungal vaccine AlgelCpG + EmEAP1 and the control vaccine AlgelCpG + Em were prepared by mixing 0.5 mL of the Em or EmEAP1 suspension (1 × 10^8^ cells/mL, prepared in saline), 9 mL of aluminum hydroxide gel (2%, *w*/*v*), and 0.5 mL of CpG solution (5 mg/mL, prepared in saline). The mixtures were gently shaken for 2 min, obtaining the vaccines AlgelCpG + EmEAP1 and AlgelCpG + Em, respectively. After being stored at 4 °C for the indicated time, the vaccines were stained by the FITC-conjugated anti-Eap1 antibody (20 mg/L), followed by observation via confocal microscopy. The stored vaccines were also sampled and centrifuged at 12,000 rpm for 10 min. The mCherry fluorescence intensity of the supernatants was determined using the fluorescence microplate reader. 

### 2.4. In Vivo Mouse Imaging of the Vaccines

All animal experiments were approved by the Animal Care and Use Committee at Nankai University (approval number 2022-SYDWLL-000397). For in vivo mouse imaging, the vaccines were prepared using Cy5-labeled synthetic cells or the Cy5-labeled Eap1 protein (final concentration in the vaccines at 1 mg/mL) with the same procedure as described above. The three kinds of vaccines, including AlgelCpG + Cy5-Eap1, AlgelCpG + Cy5-Em, and AlgelCpG + Cy5-EmEAP1, were subcutaneously injected into the hind footpad of the mice at a dose of 100 μL vaccine per mouse. The mice were anesthetized using ether at different time points after injection. The distribution of the Cy5 fluorescence in the mice was detected by a small animal fluorescence imaging system (Xenogen, Corp., Alameda, CA, USA). 

### 2.5. Lymph Node Weighing and Dendritic Cell Assays

To investigate the effect of the vaccine on lymph node growth and dendritic cell accumulation, the mice were immunized with the three kinds of vaccines (100 μL vaccine per mouse) with the prime-boost procedure on day 0 and day 7. The mice were euthanized by CO_2_ on day 14 [32]. Both the inguinal and cervical lymph nodes were sampled from the mice and weighed. The lymph node tissues were ground, and the total lymph node cells were obtained by filtering with 100-mesh sieves to remove cell aggregates. The cells were then stained by the FITC-conjugated CD11c antibody. The percent of CD11c-positive dendritic cells was measured using flow cytometry (Becton, Dickinson and Company, Franklin Lakes, NJ, USA).

### 2.6. Splenocyte Assays

The vaccinated mice were euthanized on day 14 as described above and the spleens of the mice were sampled. The splenocytes were also prepared by grinding of the spleen tissues and filtering to remove aggregates. The obtained splenocytes were stained by the FITC-conjugated anti-CD4 antibody together with the APC-conjugated anti-mouse IL-2 antibody or the APC-conjugated anti-mouse IFN-γ antibody. The cells were then examined using flow cytometry [33].

### 2.7. ELISA Assays

For determination of the serum anti-Eap1 IgG levels, venous blood was sampled from the tail veins of the vaccinated mice on day 14, day 28, and day 56. After being stored at 4 °C for 12 h, the blood was centrifuged at 1500 rpm for 5 min, obtaining the serum for ELISA assays [34]. The serum was diluted by PBS at a two-fold gradient and the diluted solutions were added into 96-well plates which were pre-incubated with the Eap1 solution (100 μg/mL). The bound IgG in the wells was further detected by the HRP-conjugated anti-mouse IgG antibody, followed by colorization via ELISA colorization agents to reflect the levels of the serum-specific IgG.

### 2.8. Mouse Vaccination and Systemic Infection

To evaluate the protective effect of the vaccines, the mice in each group (*n* = 13) were immunized with the vaccines (100 μL vaccine/mouse) via the prime-boost procedure on day 14 and 7. The immunized mice and the control mice (receiving no vaccination) were then infected by the fresh *C. albicans* SC5314 cells with intravenous inoculation at the tail vein on day 0. The survival of the infected mice was recorded from day 0 to day 14. On day 3, the kidneys of three mice in each group were sampled and weighed. The kidneys were further broken by Dounce homogenizers, and the numbers of the fungal cells in the obtained homogenates were determined using colony-forming unit assays on plates of yeast extract-peptone-dextrose medium. The TNF-α levels in the homogenate were further determined using the mouse TNF-α assay kits [35].

### 2.9. Statistical Analysis

Most of the experiments were performed with three replicates (*n* = 3), except the experiment of mouse vaccination and systemic infection (*n* = 13). The data were shown with the means ± the standard errors. The differences between the groups were analyzed by ANOVA test and Student’s *t*-test (*p* < 0.05) via SPSS 22 software (IBM, Armonk, NY, USA).

## 3. Results

### 3.1. The Synthetic EmEAP1 Cells Strongly Expose Eap1 on the Cell Surface

To construct the synthetic strain exposing the *C. albicans* Eap1, the artificial protein mChEap1 was designed by fusing the N-terminally secretory signal peptide of OmpA [36], the fluorescence protein mCherry, the Eap1 protein [30], and the C-terminally transmembrane region (CTR) [37,38]. Correspondingly, the artificial gene encoding mChEap1 (i.e., *OmpA-mCherry-EAP1-CTR*) was designed using the *E. coli*-optimized codon and de novo synthesized and cloned into the pET-28a plasmid. The plasmid was then transformed into the *E. coli* BL21 strain, obtaining the synthetic strain EmEAP1. As a control strain, the BL21 strain was also transformed by the pET-28a plasmid containing the *OmpA-mCherry-CTR*, obtaining the control strain Em. After induction by IPTG, both EmEAP1 and Em were expected to highly express and expose the synthetic proteins, respectively (Figure 1a). 

The expression and exposure of the synthetic proteins were evaluated by confocal microscopy after staining of the strains with DAPI and the FITC-conjugated anti-Eap1 antibody. As shown in Figure 1b, both of the strains emitted strong red fluorescence of mCherry, indicating high expression of the synthetic proteins in the strains. However, while the control Em cells had no obvious FITC fluorescence, the EmEAP1 cells exhibited high-level FITC fluorescence, which is mostly co-localized with the mCherry fluorescence. Fluorescence quantification by the Image J 1.53e software further confirmed that the EmEAP1 cells had a 27-fold higher intensity of FITC than the Em cells (Figure 1c). Therefore, the EmEAP1 cells successfully expressed and exposed high levels of the synthetic mChEap1 on the cell surface.

### 3.2. The Synthetic Cell-Containing Vaccine AlgelCpG + EmEAP1 Exhibits High Stability

Aluminum hydroxide gel (Algel) is one of the most important adjuvants for vaccine preparation [39,40], and the inclusion of the CpG further enhanced the immune response of the host [41,42]. For the preparation of highly efficient fungal vaccines, the synthetic cells were mixed with both Algel and CpG, obtaining the final control vaccine AlgelCpG + Em and the aimed vaccine AlgelCpG + EmEAP1, respectively (Figure 2a). The stability of the vaccines was then evaluated during different storage times. As revealed by confocal microscopy, after 28 days of storage, the control vaccine failed to bind the FITC-conjugated anti-Eap1 antibody, while the AlgelCpG + EmEAP1 vaccine could be strongly stained by the antibody, with the FITC fluorescence co-localizing with the mCherry fluorescence (Figure 2b). Moreover, during storage from 0 to 112 days, the control vaccine drastically released the mCherry protein into the extra-gel solution, while the AlgelCpG + EmEAP1 vaccine only released quite low levels of mCherry. For instance, after 112 days of storage, AlgelCpG + EmEAP1 released 1.79-fold lower levels of the fluorescence protein than AlgelCpG + Em (Figure 2c), indicating that AlgelCpG + EmEAP1 had high stability and maintained the persistence of mCherry fluorescent protein and Eap 1 in synthetic cells. In addition, cell viability assays showed that both AlgelCpG + Em and AlgelCpG + EmEAP1 had no obvious impact on the viability of the DC2.4 dendritic cells even at a concentration of 200 mg/L (Appendix A), suggesting good biocompatibility of the synthesized vaccines. Together, these results revealed that the AlgelCpG + EmEAP1 vaccine had a higher stability than the control vaccine, which may be attributed to the existence of the adhesin Eap1 on the surface of the synthetic EmEAP1 cells that stabilized the gel system.

### 3.3. The AlgelCpG + EmEAP1 Vaccine Has a Great Impact on Lymph Node Growth

The in vivo dynamic of the vaccines was then evaluated by mouse imaging. For sensitive in vivo imaging, the synthetic cells were first labeled with Cy5 for further preparation of the vaccines. As another control vaccine, the standard Cy5-labeled Eap1 protein was also used as the antigen to prepare the protein vaccine AlgelCpG + Cy5-Eap1. The vaccines (i.e., AlgelCpG + Cy5-Eap1, AlgelCpG + Cy5-Em, and AlgelCpG + Cy5-EmEAP) were subcutaneously injected into the hind footpad of the mice, and then their ability to target and stay at the lymph nodes were examined. One hour after injection, the Cy5 fluorescence in both the AlgelCpG + Cy5-EAP1 group and the AlgelCpG + Cy5-Em rapidly distributed to the inguinal and cervical lymph nodes, while the fluorescence in the AlgelCpG + Cy5-EmEAP1 group mainly distributed in the popliteal lymph nodes (Figure 3a). Moreover, 48 h after injection, the fluorescence of AlgelCpG + Cy5-EmEAP1 was greatly reduced, which may be attributed to the rapid removal of the protein antigens by the body’s immune system. On the contrary, the Cy5 fluorescence levels in the AlgelCpG + Cy5-EmEAP1 group remained high, with the fluorescence mainly accumulating at the cervical lymph nodes (Figure 3a), indicating that the synthetic cell EmEAP1 strongly enhanced antigen uptake by the lymph nodes. In addition, Cy5 fluorescence levels were significantly reduced in the AlgelCpG + Cy5-Em group, with only low levels of the fluorescence distributed in the inguinal and cervical lymph nodes. These results implied that the EmEAP1 cells had a much higher capacity to stay in the lymph nodes than the Em cells.

The improvement in antigen capture is correlated with lymph node growth and the accumulation of immune cells in the lymph nodes [43,44]. Among the four groups, the AlgelCpG + EmEAP1 group had the heaviest inguinal lymph nodes weights of the mice on day 14, which was 1.6-fold that in the mice of the AlgelCpG + Eap1 group (Figure 3b). It is speculated that the Eap1 antigen alone has a short duration and poor resident effect, whereas Eap1 exposed on the surface of EmEAP1 could efficiently reside in the lymph nodes for the induction of lymph node growth. Moreover, flow cytometry was used to detect the activation of the dendritic cells in the lymph nodes. The results showed that the AlgelCpG + EmEAP1 vaccine strongly increased the percent of CD11c^+^ dendritic cells in the lymph cells from 4% to 18%, while the two control vaccines only increased the percent to 10–13% (Figure 3c). Therefore, the AlgelCpG + EmEAP1 vaccine strongly promoted lymph node growth and induced the recruitment of dendritic cells in the lymph nodes. 

### 3.4. The AlgelCpG + EmEAP1 Vaccine Strongly Activates Splenocyte T Cells and IgG Production

During humoral immunity activated by vaccines, dendritic cells interact with CD4^+^ T cells for antigen presentation and induce the production of cytokines (such as IL-2, IFN-γ) in CD4^+^ T cells to activate the splenocyte T cells [45,46]. On day 14, after the prime-boost vaccination on day 0 and day 7, the splenocytes of each group were isolated from the mouse spleens and analyzed by flow cytometry. The AlgelCpG + EmEAP1 group had a higher percentage of CD4^+^ T cells in the splenocytes compared to the other three groups (3.5% versus 1.5~2.7%, Figure 4a). Meanwhile, in terms of activated splenocytes CD4^+^ T cells, the AlgelCpG + EmEAP1 group also had the highest activated CD4^+^ T cells producing IL-2^+^ and IFN-γ^+^ immune factors (Figure 4b,c). Overall, these results suggest that AlgelCpG + EmEAP1 has the strongest activity to recruit and activate CD4^+^ T cells in the spleen.

To further verify the effect of the synthetic vaccines on the humoral immune response, the serum anti-Eap1 IgG levels in the vaccinated mice were detected using ELISA assays on days 14, 28, and 56, respectively (Figure 4d). At each tested timepoint, the AlgelCpG + EmEAP1 group exhibited much higher IgG levels than the other three groups. For example, 28 days after injection, the titer of the serum-specific IgG in the AlgelCpG + EmEAP1 group was >60,000, while the titer in the AlgelCpG + Eap1 group was <20,000, and the titer in the AlgelCpG + Em group together with the control group were almost undetectable. Even after 56 days, the AlgelCpG + EmEAP1 retained high levels of the specific IgG at the titer of 60,000 (Figure 4d). Therefore, The AlgelCpG + EmEAP1 group had the highest ability to induce long-term production of the anti-Eap1 IgG after the prime-boost vaccination.

### 3.5. The AlgelCpG + EmEAP1 Vaccine Protects the Mice from Systemic Fungal Infection

In order to investigate the protective effect of the vaccines against systemic fungal infections, the mice were firstly vaccinated with the prime-boost procedure on day 14 and day 7, and then injected with fresh *C. albicans* cells via the tail vein on day 0 for survival assays. After the *C. albicans* infection, all of the unvaccinated control mice died within 5 days, and all the mice receiving the vaccination of AlgelCpG + Em or AlgelCpG + Eap1 died within 6 and 9 days, respectively (Figure 5a), indicating that both Em and Eap1 failed to efficiently protect the mice from systemic infection. Remarkably, the death of the mice receiving the vaccination of AlgelCpG + EmEAP1 was drastically prolonged, with 70% of the mice still alive after 8 days. Moreover, 50% of the mice in the AlgelCpG + EmEAP1 group remained alive even after 14 days (Figure 5a). Therefore, the AlgelCpG + EmEAP1 vaccine efficiently prolonged the survival time of the infected mice and protected half of the mice from death induced by systemic fungal infection.

The kidney is the primary site of systemic infection by *C. albicans*, and its fungal burden could reflect the degree of systemic infection by this pathogen [47,48]. By weighing the kidneys of the mice infected with *C. albicans* on day 3, it was found that the AlgelCpG + EmEAP1 group had the lowest kidney weight (Figure 5b), indicating the lowest degree of inflammation-related kidney enlargement induced by *C. albicans* in this group. According to the measurement of fungal burden, mice vaccinated with AlgelCpG + EmEAP1 had much lower fungal burden as compared to the mice vaccinated with AlgelCpG + Eap1 or AlgelCpG + Em (Figure 5c), suggesting that this vaccine induced a remarkable immune response to inhibit kidney infection of the fungal pathogen.

Tumor necrosis factor α (TNF-α) is a pro-inflammatory cytokine that plays a key role in pathogen-induced inflammatory response [49]. The expression of the cytokine TNF-α was found to be lower in the kidneys of the AlgelCpG + EmEAP1 group than in the kidneys of the other groups, indicating that the kidneys of this group had less inflammation (Figure 5d). This is consistent with the result of kidney weighing. Thus, the vaccines containing EmEAP1 effectively activated the immune response and protected the mice from life-threatening fungal infections.

## 4. Discussion

Owing to the severe situation of antibiotic abuse and immune deficiency in clinics, systemic fungal infections are becoming one of the most important causes of human death [50,51]. The development of efficient, economic, and safe fungal vaccines to protect individuals from these dangerous infections is an economic strategy for the attenuation of fungal threats. As compared to previous fungal vaccines that needed complicated preparation of polysaccharide groups or protein fragments, or needed living fungal cells with attenuated but remained virulence, the vaccine prepared on the basis of synthetic biology techniques in this study only needed preparation of the synthetic bacterial cells by convenient fermentation, followed by easy mixing of the bacterial cells with aluminum hydroxide gel and CpG. The obtained vaccine exhibited high biosafety and an efficient ability to protect mice from systemic fungal infection. In the future, it is no doubt that synthetic biology techniques may further promote the development of novel fungal vaccines.

It is interesting that the AlgelCpG + EmEAP1 vaccine had a higher stability than the control AlgelCpG + Em vaccine, although both of the vaccines had *E. coli* cells and aluminium hydroxide gels. The mechanism of enhanced stability may be attributed to the high affinity of the adhesin Eap1 on the surface of the synthetic cells to the adjuvant components. In *C. albicans*, this adhesin mediates the binding of the fungal cells to both the polystyrene surface and the epithelial cell surface. Its mutation therefore leads to a remarkable decrease in adhesion of the fungal cells to these surfaces [52]. It is anticipated that Eap1 exposed by the synthetic cells may also strongly bind the groups of the aluminum hydroxide particles, functioning as the bridge between the particles for forming a stable gel network. Moreover, the adhesin may also self-assemble to form a protective layer on the cell surface, protecting the antigens released from the synthetic cells into the extra-gel solution. Consequently, the AlgelCpG + EmEAP1 vaccine exhibited higher stability than the Eap1-free AlgelCpG + Em vaccine during long-term storage.

In recent years, fungal extracellular vesicles carrying immunogenic proteins, such as mannan protein MP88, chitin deacetylase Cda family proteins, and Vep proteins, have been used for vaccine preparation against fungal pathogens such as *Candida* and *Paracoccidioides* [53,54]. However, due to the fact that fungal extracellular vesicles are a mixture of multiple components, some of which may have toxic effects, there are obstacles to developing vesicle-based vaccines for application [55]. In previous studies of *C. albicans* vaccines, a series of fungal cell surface proteins (e.g., Sap2, Als3, Hsp90, and Hyr1) were used as antigens for vaccine development [20,21]. In this study, we further used the novel cell surface protein Eap1 as the basis of the synthetic cell construction. This protein was chosen owing to its high expression and surface exposure on both yeast and hyphal *C. albicans* cells, and its property of adhesin that may further mediate the assembly of the adjuvants on the cell surface of the synthetic cells. It was reported that the assembly of adjuvants may facilitate the recruitment of immune cells around the vaccine components and further activation of the humoral immunity pathway [56,57]. Herein, we found that the synthetic vaccine containing the Eap1-exposed synthetic cells exhibited a much higher capacity to activate the CD4^+^ T cells and to improve the level of specific IgG levels, which may be associated with the assembly-inducing effect of the synthetic cells.

In the in vivo imaging experiments, we found that the Cy5 fluorescence of the AlgelCpG + Cy5-EmEAP1 vaccine had a more prolonged residence time than both AlgelCpG + Cy5-EAP1 and AlgelCpG + Cy5-Eap1. It is easy to understand that the free Eap1, owing to its small size, could rapidly pass through the lymph nodes and be removed by the body via the circulation system. In terms of Cy5-EmEAP1 and Cy5-Em, while the fluorescence of Cy5-Em remarkably attenuated after 48 h, the fluorescence of Cy5-EmEAP1 remained at high levels in the lymph nodes (Figure 3a). A possible explanation is that the exposed Eap1 induced rapid accumulation of dendritic cells in the lymph nodes, forming aggregates in the lymph nodes. The crosslinking effect of the adhesin Eap1 further led to the high stability of the EmEAP1-dendritic cell aggregates and consequent long-term residence of the synthetic cells. The enhanced residence of the antigen-burden cells may further contribute to the persistent antigen presentation and activation of downstream immune cells for strengthened humoral immunity.

By measuring kidney fungal burden, kidney TNF-α levels, and kidney weights on day 3, we found that the fungal burden was proportional to the TNF-α levels and kidney weights; that is, the higher fungal numbers, the higher the inflammatory factor levels and kidney weights. This indicates that the vaccine could attenuate fungal infection and colonization in the kidneys, thereby alleviating the inflammatory response in the kidneys (Figure 5b–d). This may be because the AlgelCpG + Cy5-EmEAP1 vaccine evoked high levels of the fungus-specific IgG, leading to strong humoral immunity against the fungal cells invading the kidneys. Consequently, the mice that received the vaccination of AlgelCpG + Cy5-EmEAP1 exhibited much weaker kidney inflammation than the control mice, which suffered from severe fungal infections in the kidneys. 

Herein, we used the *C. albicans* adhesin Eap1 as the aimed antigen for the prevention of systemic infection caused by this pathogen. It is known that adhesins are important virulence factors of fungal pathogens, which mediate the effective adhesion of the fungal pathogens to the host cell surface and further invasion into the host cells [58]. For example, the fungal pathogen *A. fumigatus* possesses the adhesin Mp1 on the cell surface [59]. In the future, this adhesin could also be used for the design of artificial proteins and the construction of synthetic cells, which may facilitate the development of fungal vaccines against systemic infections induced by other kinds of dangerous fungal pathogens.

In this study, we showed that AlgelCpG + EmEAP1 remained stable and resided in the lymph nodes of mice 112 days after injection. However, the long-term durability of the vaccine and its efficiency in protecting patients from recurrent and multi-drug resistant *Candida albicans* remains to be investigated. In the meantime, it is suggested to further improve the capacity of the vaccine to reduce the fungal burden and the chance of infection in patients. In recent years, the development of new antifungal therapeutics has remained slow [17]. Therefore, our further investigations will focus on the promotion of the efficiency of fungal vaccines for real clinical application.

## 5. Conclusions

In conclusion, this study developed a novel synthetic cell-based fungal vaccine for the protection of the body from life-threatening systemic fungal infections using synthetic biology techniques. The synthetic cell EmEAP1 was constructed by the design of the artificial protein fusing the *E. coli* OmpA signal peptide, the fluorescence protein mCherry, the *C. albicans* adhesin Eap1, and the *E. coli* CTR region, followed by de novo gene synthesis and transformation of the gene into the *E. coli* chassis. The synthetic vaccine was then prepared by mixing the EmEAP1 cells with aluminum hydroxide gel and the TLR9 agonist CpG. The prepared vaccine exhibited high stability during long-term storage and abundantly resided in the lymph nodes for the induction of lymph node growth and recruitment of the CD11c^+^ dendritic cells. After immunization of the mice via the prime-boost procedure, the synthetic fungal vaccine strongly activated the CD4^+^ T splenocytes and induced high levels of anti-Eap1 antibodies in the serum. In a systemic fungal infection model, immunization of the vaccines drastically prolonged the survival time of the infected mice, inhibited kidney colonization of the fungal pathogens, and attenuated the levels of the kidney inflammatory factor TNF-α. This study sheds light on the application of synthetic biology techniques in the development of efficient fungal vaccines.

## Figures and Tables

**Figure 1 vaccines-11-01521-f001:**
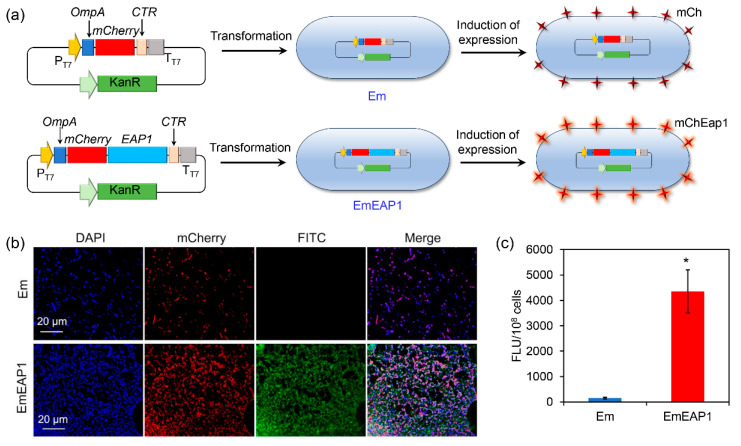
Schematic illustration (**a**) and characterization of the synthetic *E. coli* strains Em and EmEAP1. (**a**) A scheme indicating the construction of the strains and the induction of protein exposure in the synthetic cells. (**b**) Confocal images of the Em and EmEAP1 cells after induction of IPTG for 12 h. The cells were stained by DAPI (blue fluorescence, indicating bacterial DNA) and the FITC (green fluorescence)-conjugated anti-Eap1 body for observation. The mCherry field indicates the synthetic mCh/mChEap1 expressed by the *E. coli* cells. (**c**) Quantification of the FITC fluorescence intensity of the stained cells. The asterisk indicates a significant difference between the two groups (*p* < 0.05).

**Figure 2 vaccines-11-01521-f002:**
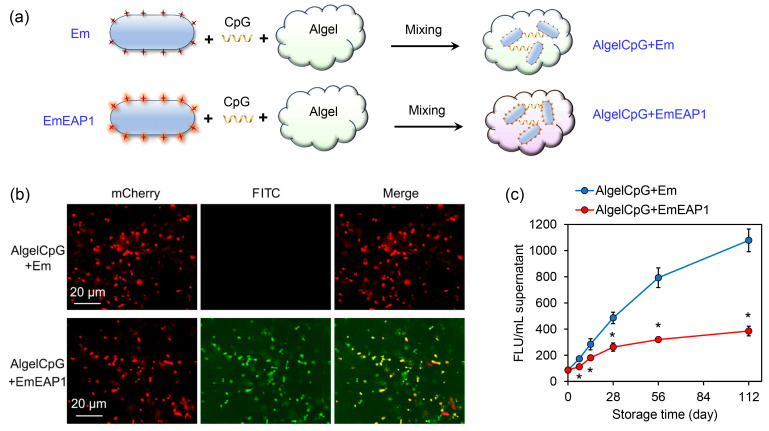
Preparation protocol and characterization of the AlgelCpG + Em and AlgelCpG + EmEAP1 vaccines. (**a**) A scheme illustrating the preparation of the gel vaccines. (**b**) Confocal images of the gels after 28 days of storage. The mCherry and FITC fields indicate the synthetic mCh/mChEap1 protein and FITC-conjugated anti-Eap1 antibody, respectively. (**c**) Fluorescence intensity of the released supernatants by centrifugation of the gels after storing for the indicated time. The FLU values of the supernatant were detected by a fluorescent microplate reader. The asterisks indicate a significant difference between the two groups at each time point (*p* < 0.05).

**Figure 3 vaccines-11-01521-f003:**
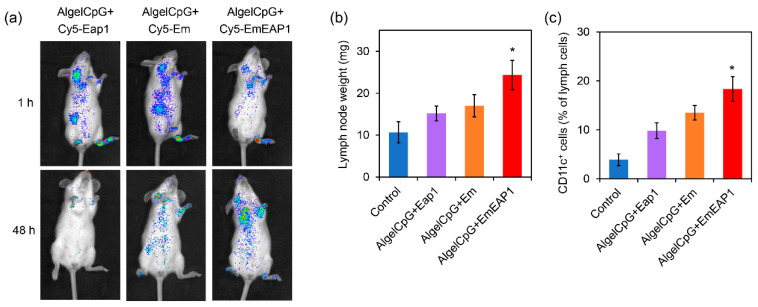
Imaging of the Cy 5-labeled antigen Eap1. (**a**) Alteration of lymph node weights (**b**) and change of CD11c^+^ cells (**c**) in the mice receiving vaccination. (**a**) Imaging of the mice receiving the vaccines with Cy5-labeled Eap1 or synthetic cells. The fluorescence indicates the distribution of Cy5-labeled Eap1. (**b**) Weight of the inguinal and cervical lymph nodes on day 14 in the mice receiving the prime-boost vaccination on day 0 and day 7. (**c**) The percent of CD11c^+^ dendritic cells in the inguinal and cervical lymph nodes. The asterisks indicate a significant difference between the AlgelCpG + EmEAP1 group and the other groups (*p* < 0.05).

**Figure 4 vaccines-11-01521-f004:**
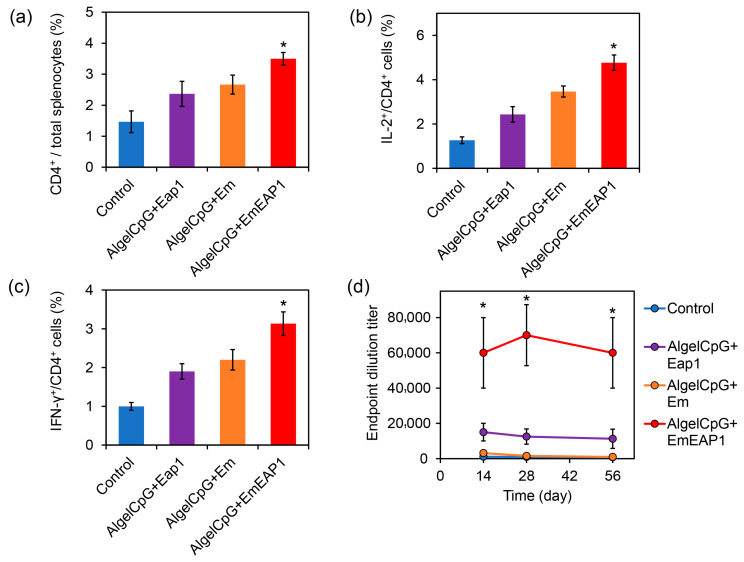
Activation of splenocyte T cells and induction of anti-Eap1 IgG production by the vaccines. (**a**) The percentage of CD4^+^ cells in the splenocytes. The mice were immunized by the vaccines via the prime-boost procedure on day 0 and day 7, and killed on day 14. The spleens were sampled for immunostaining and flow cytometry. (**b**) The percent of IL-2^+^ cells in the CD4^+^ cells. (**c**) The percent of IFN-γ^+^ cells in the CD4^+^ cells. (**d**) Serum anti-Eap1 IgG levels at different time points. The asterisks indicate a significant difference between the AlgelCpG + EmEAP1 group and the other groups (*p* < 0.05).

**Figure 5 vaccines-11-01521-f005:**
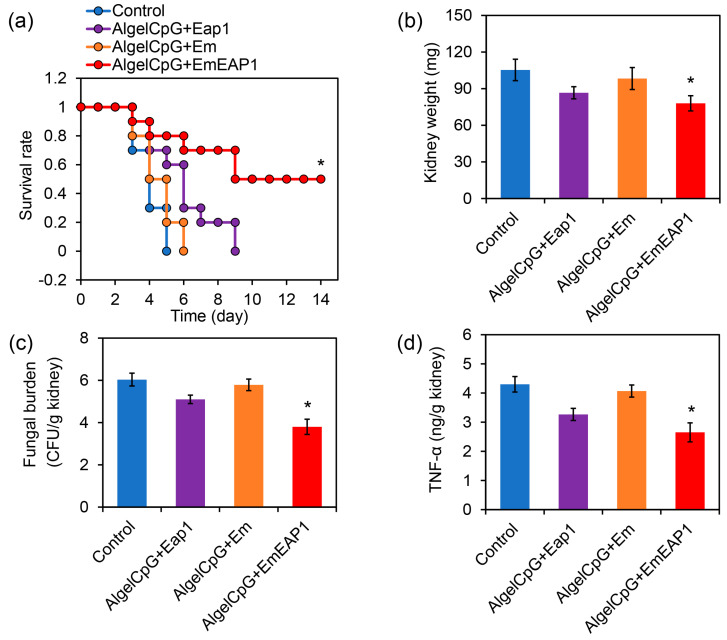
Protection of the mice from *C. albicans* systemic infection by the vaccines. (**a**) Survival curves of the mice. The mice were vaccinated by the vaccines or by PBS (Control) with the prime-boost procedure on day 14 and day 7. The mice were then infected by the fungal cells (5 × 10^6^ cells/mouse) via tail intravenous injection on day 0, followed by survival monitoring. (**b**) Kidney weight of the mice on day 3. (**c**) Fungal burden in the kidneys on day 3. (**d**) TNF-α levels in the kidneys on day 3. The asterisks indicate a significant difference between the AlgelCpG + EmEAP1 group and the other groups (*p* < 0.05).

## Data Availability

Not applicable.

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
