# Peer review of "Construction of *Candida albicans* Adhesin-Exposed Synthetic Cells for Preventing Systemic Fungal Infection"

_vaccines, 2023, doi:10.3390/vaccines11101521_

Round 1

Reviewer 1 Report

Dear authors,

greetings!

1 Construction of Candida albicans adhesin-exposed synthetic 2 cells for vaccination against systemic fungal infection-Please modify

2In this study, we prepared a synthetic cell-based fungal vaccine for prevent- 12 ing systemic fungal infections by synthetic biology techniques. The synthetic cell EmEAP1 was con- 13 structed by transformation of the Escherichia coli chassis by a de novo synthetic fragment encoding 14 the protein mChEap1 composed of the OmpA peptide, mCherry, the Candida albicans adhesin Eap1, 15 and the C-terminally transmembrane region. The EmEAP1 cells highly exposed the artificial protein 16 on cell surface under IPTG induction. (please provide updated information,use italics for all microorganisms)

3In this study, we prepared a synthetic cell-based fungal vaccine for prevent- 12 ing systemic fungal infections by synthetic biology techniques. The synthetic cell EmEAP1 was con- 13 structed by transformation of the Escherichia coli chassis by a de novo synthetic fragment encoding 14 the protein mChEap1 composed of the OmpA peptide, mCherry, the Candida albicans adhesin Eap1, 15 and the C-terminally transmembrane region. The EmEAP1 cells highly exposed the artificial protein 16 on cell surface under IPTG induction. iTs not clear

4With the increased number of immune-compromised population induced by virus 30 infection, radiotherapy or chemotherapy, the abuse of antibiotics, and the emerging cases 31 of organ transplantation, human beings are suffering from increasing risks of life-threat- 32 ening fungal infections [1-3]. In clinic, the prevalent fungal pathogens, e.g., Candida albi- 33 cans, Candida auris, Aspergillus fumigatus, Cryptococcus neoformans, etc. Especially, Candida- 34 related infections, which are frequently associated with candidemia and multi-organ 35 damages, contribute to about 40% of hospital acquired fungal infections with high mor- 36 tality [4-7]. Up to now, unfortunately, there are only a few kinds of antifungal drugs, such 37 as azoles, amphotericin, and echinocandins, that are commonly used for treatment( of 38 these dangerous infections [8].(please modify)

5Materials and Methods 73 2.1. Materials 74 The aluminium hydroxide gel was purchased from Aladdin, China. The TLR9 ag- 75 onist CpG 1826 (5′-TCCATGACGTTCCTGACGTT-3′) was synthesized by Sangon Bio- 76 technology, China. The E. coli strain BL21 was obtained from GenScript, China. To prepare 77 the anti-Eap1 antibody, the rabbits were immunized by the purified Eap1 protein from 78 genetically engineered E. coli cells via the prime-boost-boost procedure, followed by se- 79 rum isolation and antibody purification. The obtained antibody was further labeled by 80 FITC in 10 mM NaHCO3 solution (pH = 9.0). DAPI and FITC was purchased from Sigma, 81 USA. The Cy5 NHS ester was purchased from APExBIO, USA. The FITC-conjugated 82 CD11c and CD4 antibodies, and the APC-conjugated anti-mouse IL-2 and IFN-γ antibod- 83 ies were purchased from BioLegend, USA. The HRP-conjugated anti-mouse IgG antibody 84 was purchased from Santa Cruz, USA. The ELISA colorization agents were obtained from 85 Dingguo, China. The mouse TNF-α assay kits were purchased from Jianglaibio, China. 86 87 2.2. Construction and observation of the synthetic cells 88 To construct the synthetic strain EmEAP1, the artificial gene OmpA-mCherry-EAP1- 89 CTR was firstly designed by using the optimized codon of E. coli, based on the sequence 90 of the corresponding encoded protein mChEap1. The designed gene was then de novo 91 synthesized by GenScript, China, and then cloned into the plasmid pET-28a, obtaining the 92 plasmid pET-OmpA-mCherry-EAP1-CTR, in which the OmpA-mCherry-EAP1-CTR gene 93 was under the control of the strong T7 promoter and the T7 terminator. The plasmid was 94 then transformed into the BL21 strain, obtaining the synthetic E. coli strain EmEAP1. As 95 the control strain, Em was constructed by transformation of the BL21 strain with the plas- 96 mid pET- OmpA-mCherry-CTR, in which the EAP1 sequence is absent. Both Em and 97 EmEAP1 cells were cultured in the liquid LB medium containing 0.5 mM IPTG for 12 h, 98 Vaccines 2023, 11, x FOR PEER REVIEW 3 of 14 followed by staining with DAPI (10 mg/L) and the FITC-conjugated anti-Eap1 antibody 99 (20 mg/L) for 20 min. The stained cells were then observed by confocal microscopy 100 (FV1000, Olympus). The FITC fluorescence of the stained cells were determined by using 101 a fluorescence microplate reader (Enspire, PerkinElmer, USA). 102 103 2.3. Preparation and observation of the fungal vaccines 104 The fungal vaccine AlgelCpG+EmEAP1 and the control vaccine AlgelCpG+Em were 105 prepared by mixing 0.5 mL of the Em or EmEAP1 suspension (1×108 cells/mL, prepared 106 in saline), 9 mL of aluminium hydroxide gel (2%, W/V), and 0.5 mL of CpG solution (5 107 mg/mL, prepared in saline). The mixtures were gently shaken for 2 min, obtaining the 108 vaccines AlgelCpG+Em and AlgelCpG+EmEAP1, respectively. After storage of the vac- 109 cines at 4 °C for indicated time, the vaccines were stained by the FITC-conjugated anti- 110 Eap1 antibody (20 mg/L), followed by observation via the confocal microscopy. The stor- 111 age vaccines were also sampled and centrifuged at 12, 000 rpm for 10 min. The mCherry 112 fluorescence intensity of the supernatants were determined by using the fluorescence mi- 113 croplate reader.(major revision is required)

64. In vivo mouse imaging of the vaccines 116 All animal experiments were approved by the Animal Care and Use Committee at 117 Nankai University (Approval number 2022-SYDWLL-000397). For in vivo mouse imag- 118 ing, the vaccines were prepared by using the Cy5-labeled synthetic cells or the Cy5-la- 119 beled Eap1 protein (final concentration of in the vaccines at 1 mg/mL) with the same pro- 120 cedure as described above. The three kinds of vaccines, including AlgelCpG+Cy5-Eap1, 121 AlgelCpG+Cy5-Em, AlgelCpG+Cy5-EmEAP1, were subcutaneously injected into the hind 122 footpad of the mice at the dose of 100 μL vaccine/mouse. At the different time points after 123 injection, the mice were anesthetized by using ether, and the distribution of the Cy5 fluo- 124 rescence in the mice were detected by a small-animal fluorescence imaging system (Xeno- 125 gen, USA). 126 127 2.5. Lymph node weighing and dendritic cell assays 128 To investigate the effect of the vaccine on lymph node growth and dendritic cell ac- 129 cumulation, the mice were immunized by the three kinds of vaccines (100 μL vac- 130 cine/mouse) with the prime-boost procedure on day 0 and day 7. The mice were eu- 131 thanized by CO2 on day 14. Both the inguinal and cervical lymph nodes were sampled 132 from the mice and weighed. The total cells of lymph nodes were obtained by grinding the 133 lymph node tissues and filtering with 100-mesh sieves. The cells were then stained by the 134 FITC-conjugated CD11c antibody, and then the CD11c-positive dendritic cells were ex- 135 amined by using flow cytometry (BD, USA)(please revise)

6Splenocyte assays 138 The vaccinated mice were euthanized on day 14 as described above, and the 139 spleens of the mice were sampled. The splenocytes were prepared also by grinding and 140 filtering. The obtained splenocytes were stained by the FITC-conjugated CD4 antibody 141 and the APC-conjugated anti-mouse IL-2 and IFN-γ antibodies. The cells were then ex- 142 amined by using flow cytometry [30]. 143 144 2.7. ELISA assays 145 For determination of the serum anti-Eap1 IgG levels, the venous blood was sampled 146 from the tail veins of the vaccinated mice on day 14, day 28 and day 56. After storage at 4 147 °C for 12 h, the blood was centrifuged at 1500 rpm for 5 min, obtaining the serum for 148 ELISA assays. The serum was diluted by PBS at a two-fold gradient, and the diluted so- 149 lutions were( added into the 96-well plates pre-incubated with Eap1 solution (100 μg/mL). 150 The bound IgG in the wells was further incubated by the HRP-conjugated anti-mouse IgG 1(its not clear please modify the above information )

7 please check plagiarism)

8 please mention methodology clearly

9 Please check plagiarism

10 Please arrange references as per medeley and zotero

11 please follow author guidelines

Best regards

Please check with grammarly

paraphrasing check

minor corrections needed in methodoloy and introduction

Author Response

Response to Reviewer 1

Dear authors,

greetings!

1 Construction of Candida albicans adhesin-exposed synthetic 2 cells for vaccination against systemic fungal infection-Please modify.

Response:

As you kindly suggested, the title of this paper has been modified (Page 1, Lines 2-3):

“Construction of Candida albicans adhesin-exposed synthetic cells for preventing systemic fungal infection”

2 In this study, we prepared a synthetic cell-based fungal vaccine for prevent- 12 ing systemic fungal infections by synthetic biology techniques. The synthetic cell EmEAP1 was con- 13 structed by transformation of the Escherichia coli chassis by a de novo synthetic fragment encoding 14 the protein mChEap1 composed of the OmpA peptide, mCherry, the Candida albicans adhesin Eap1, 15 and the C-terminally transmembrane region. The EmEAP1 cells highly exposed the artificial protein 16 on cell surface under IPTG induction. (please provide updated information, use italics for all microorganisms)

Response:

The information has been updated, and the name of the microorganisms have been written with the italic form (Page 1, Lines 17-21):

“The synthetic cell EmEAP1 was constructed by transformation of the Escherichia coli chassis by a de novo synthetic fragment encoding the protein mChEap1 composed of the E. coli OmpA peptide, the fluorescence protein mCherry, the Candida albicans adhesin Eap1, and the C-terminally transmembrane region. The EmEAP1 cells highly exposed mChEap1 on cell surface under IPTG induction.”

3 In this study, we prepared a synthetic cell-based fungal vaccine for prevent- 12 ing systemic fungal infections by synthetic biology techniques. The synthetic cell EmEAP1 was con- 13 structed by transformation of the Escherichia coli chassis by a de novo synthetic fragment encoding 14 the protein mChEap1 composed of the OmpA peptide, mCherry, the Candida albicans adhesin Eap1, 15 and the C-terminally transmembrane region. The EmEAP1 cells highly exposed the artificial protein 16 on cell surface under IPTG induction. iTs not clear

Response:

  The sentence has been revised for accuracy (Page 1, Lines 16-21):

“In this study, we prepared a synthetic cell-based fungal vaccine for preventing systemic fungal infections by synthetic biology techniques. The synthetic cell EmEAP1 was constructed by transformation of the Escherichia coli chassis by a de novo synthetic fragment encoding the protein mChEap1 composed of the E. coli OmpA peptide, the fluorescence protein mCherry, the Candida albicans adhesin Eap1, and the C-terminally transmembrane region. The EmEAP1 cells highly exposed mChEap1 on cell surface under IPTG induction.”

4 With the increased number of immune-compromised population induced by virus 30 infection, radiotherapy or chemotherapy, the abuse of antibiotics, and the emerging cases 31 of organ transplantation, human beings are suffering from increasing risks of life-threat- 32 ening fungal infections [1-3]. In clinic, the prevalent fungal pathogens, e.g., Candida albi- 33 cans, Candida auris, Aspergillus fumigatus, Cryptococcus neoformans, etc. Especially, Candida- 34 related infections, which are frequently associated with candidemia and multi-organ 35 damages, contribute to about 40% of hospital acquired fungal infections with high mor- 36 tality [4-7]. Up to now, unfortunately, there are only a few kinds of antifungal drugs, such 37 as azoles, amphotericin, and echinocandins, that are commonly used for treatment( of 38 these dangerous infections [8].(please modify)

Response:

The Introduction part has been revised (Page 1, Lines 34-45):

“With the increased number of immune-compromised individuals suffering from virus infection, radiotherapy/chemotherapy, antibiotic abuse, and organ transplantation, human beings are suffering from high risks of life-threatening fungal infections [1-3]. In clinic, the fungal pathogens, e.g., Candida albicans, Candida auris, Aspergillus fumigatus, Cryptococcus neoformans, etc., are becoming prevalent. Especially, Candida-related infections, which are frequently associated with candidemia and multi-organ damages, contribute to about 40% of hospital acquired fungal infections with high mortality [4-7]. Up to now, unfortunately, there are only a few kinds of antifungal drugs, such as azoles, amphotericin, and echinocandins [8]. The appearance of more and more drug-resistant Candida pathogenic strains is impacting the present antifungal therapies. These strains take the lives of millions of people every year [9,10]. Therefore, it is urgent to develop novel strategies against the fungal infections.”

5 Materials and Methods 73 2.1. Materials 74 The aluminium hydroxide gel was purchased from Aladdin, China. The TLR9 ag- 75 onist CpG 1826 (5′-TCCATGACGTTCCTGACGTT-3′) was synthesized by Sangon Bio- 76 technology, China. The E. coli strain BL21 was obtained from GenScript, China. To prepare 77 the anti-Eap1 antibody, the rabbits were immunized by the purified Eap1 protein from 78 genetically engineered E. coli cells via the prime-boost-boost procedure, followed by se- 79 rum isolation and antibody purification. The obtained antibody was further labeled by 80 FITC in 10 mM NaHCO3 solution (pH = 9.0). DAPI and FITC was purchased from Sigma, 81 USA. The Cy5 NHS ester was purchased from APExBIO, USA. The FITC-conjugated 82 CD11c and CD4 antibodies, and the APC-conjugated anti-mouse IL-2 and IFN-γ antibod- 83 ies were purchased from BioLegend, USA. The HRP-conjugated anti-mouse IgG antibody 84 was purchased from Santa Cruz, USA. The ELISA colorization agents were obtained from 85 Dingguo, China. The mouse TNF-α assay kits were purchased from Jianglaibio, China. 86 87 2.2. Construction and observation of the synthetic cells 88 To construct the synthetic strain EmEAP1, the artificial gene OmpA-mCherry-EAP1- 89 CTR was firstly designed by using the optimized codon of E. coli, based on the sequence 90 of the corresponding encoded protein mChEap1. The designed gene was then de novo 91 synthesized by GenScript, China, and then cloned into the plasmid pET-28a, obtaining the 92 plasmid pET-OmpA-mCherry-EAP1-CTR, in which the OmpA-mCherry-EAP1-CTR gene 93 was under the control of the strong T7 promoter and the T7 terminator. The plasmid was 94 then transformed into the BL21 strain, obtaining the synthetic E. coli strain EmEAP1. As 95 the control strain, Em was constructed by transformation of the BL21 strain with the plas- 96 mid pET- OmpA-mCherry-CTR, in which the EAP1 sequence is absent. Both Em and 97 EmEAP1 cells were cultured in the liquid LB medium containing 0.5 mM IPTG for 12 h, 98 Vaccines 2023, 11, x FOR PEER REVIEW 3 of 14 followed by staining with DAPI (10 mg/L) and the FITC-conjugated anti-Eap1 antibody 99 (20 mg/L) for 20 min. The stained cells were then observed by confocal microscopy 100 (FV1000, Olympus). The FITC fluorescence of the stained cells were determined by using 101 a fluorescence microplate reader (Enspire, PerkinElmer, USA). 102 103 2.3. Preparation and observation of the fungal vaccines 104 The fungal vaccine AlgelCpG+EmEAP1 and the control vaccine AlgelCpG+Em were 105 prepared by mixing 0.5 mL of the Em or EmEAP1 suspension (1×108 cells/mL, prepared 106 in saline), 9 mL of aluminium hydroxide gel (2%, W/V), and 0.5 mL of CpG solution (5 107 mg/mL, prepared in saline). The mixtures were gently shaken for 2 min, obtaining the 108 vaccines AlgelCpG+Em and AlgelCpG+EmEAP1, respectively. After storage of the vac- 109 cines at 4 °C for indicated time, the vaccines were stained by the FITC-conjugated anti- 110 Eap1 antibody (20 mg/L), followed by observation via the confocal microscopy. The stor- 111 age vaccines were also sampled and centrifuged at 12, 000 rpm for 10 min. The mCherry 112 fluorescence intensity of the supernatants were determined by using the fluorescence mi- 113 croplate reader.(major revision is required)

Response:

The description of the materials and methods has been revised (Pages 2-3, Lines 76-117):

2.1. Materials

The aluminium hydroxide gel was purchased from Aladdin, China. The TLR9 agonist CpG 1826 (5′-TCCATGACGTTCCTGACGTT-3′) was synthesized by Sangon Biotechnology, China. The E. coli strain BL21 (the chassis) was obtained from GenScript, China. To prepare the anti-Eap1 antibody, the rabbits were immunized by the purified Eap1 protein from genetically engineered E. coli cells via the prime-boost-boost procedure, followed by serum isolation and antibody purification. The obtained antibody was further labeled by FITC in 10 mM NaHCO3 solution (pH = 9.0). DAPI and FITC were purchased from Sigma, USA. The Cy5 NHS ester was purchased from APExBIO, USA. The FITC-conjugated anti-CD11c and anti-CD4 antibodies, together with the APC-conjugated anti-mouse IL-2 and IFN-γ antibodies were purchased from BioLegend, USA. The HRP-conjugated anti-mouse IgG antibody was purchased from Santa Cruz, USA. The ELISA colorization agents were obtained from Dingguo, China. The mouse TNF-α assay kits were purchased from Jianglaibio, China.

2.2. Construction and observation of the synthetic cells

To construct the synthetic strain EmEAP1, the artificial gene OmpA-mCherry-EAP1-CTR was firstly designed by using the optimized codons of E. coli, based on the sequence of the corresponding encoded protein mChEap1. The designed gene was then de novo synthesized by GenScript, China, and then cloned into the plasmid pET-28a, obtaining the plasmid pET-OmpA-mCherry-EAP1-CTR, in which the OmpA-mCherry-EAP1-CTR gene was under the control of the strong T7 promoter and the T7 terminator. The plasmid was then transformed into the BL21 strain, obtaining the synthetic E. coli strain EmEAP1. As the control strain, Em was constructed by transformation of the BL21 strain with the plasmid pET-OmpA-mCherry-CTR, in which the EAP1 sequence was absent. Both Em and EmEAP1 cells were cultured in the liquid LB medium containing 0.5 mM IPTG for 12 h, followed by staining with DAPI (10 mg/L) and the FITC-conjugated anti-Eap1 antibody (20 mg/L) for 20 min. The stained cells were then observed by confocal microscopy (FV1000, Olympus). The fluorescence intensity units (FLU) of FITC of the stained cells was determined by using a fluorescence microplate reader (Enspire, PerkinElmer, USA).

2.3. Preparation and observation of the fungal vaccines

The fungal vaccine AlgelCpG+EmEAP1 and the control vaccine AlgelCpG+Em were prepared by mixing 0.5 mL of the Em or EmEAP1 suspension (1×108 cells/mL, prepared in saline), 9 mL of the aluminium hydroxide gel (2%, W/V), and 0.5 mL of the CpG solution (5 mg/mL, prepared in saline). The mixtures were gently shaken for 2 min, obtaining the vaccines AlgelCpG+EmEAP1 and AlgelCpG+Em, respectively. After stored at 4 °C for indicated time, the vaccines were stained by the FITC-conjugated anti-Eap1 antibody (20 mg/L), followed by observation via the confocal microscopy. The stored vaccines were also sampled and centrifuged at 12, 000 rpm for 10 min. The mCherry fluorescence intensity of the supernatants were determined by using the fluorescence microplate reader.”

  1. In vivo mouse imaging of the vaccines 116 All animal experiments were approved by the Animal Care and Use Committee at 117 Nankai University (Approval number 2022-SYDWLL-000397). For in vivo mouse imag- 118 ing, the vaccines were prepared by using the Cy5-labeled synthetic cells or the Cy5-la- 119 beled Eap1 protein (final concentration of in the vaccines at 1 mg/mL) with the same pro- 120 cedure as described above. The three kinds of vaccines, including AlgelCpG+Cy5-Eap1, 121 AlgelCpG+Cy5-Em, AlgelCpG+Cy5-EmEAP1, were subcutaneously injected into the hind 122 footpad of the mice at the dose of 100 μL vaccine/mouse. At the different time points after 123 injection, the mice were anesthetized by using ether, and the distribution of the Cy5 fluo- 124 rescence in the mice were detected by a small-animal fluorescence imaging system (Xeno- 125 gen, USA). 126 127 2.5. Lymph node weighing and dendritic cell assays 128 To investigate the effect of the vaccine on lymph node growth and dendritic cell ac- 129 cumulation, the mice were immunized by the three kinds of vaccines (100 μL vac- 130 cine/mouse) with the prime-boost procedure on day 0 and day 7. The mice were eu- 131 thanized by CO2 on day 14. Both the inguinal and cervical lymph nodes were sampled 132 from the mice and weighed. The total cells of lymph nodes were obtained by grinding the 133 lymph node tissues and filtering with 100-mesh sieves. The cells were then stained by the 134 FITC-conjugated CD11c antibody, and then the CD11c-positive dendritic cells were ex- 135 amined by using flow cytometry (BD, USA)(please revise)

Response:

These method sections have been revised (Pages 3-4, Lines 119-139):

2.4. In vivo mouse imaging of the vaccines

All animal experiments were approved by the Animal Care and Use Committee at Nankai University (Approval number 2022-SYDWLL-000397). For in vivo mouse imaging, the vaccines were prepared by using the Cy5-labeled synthetic cells or the Cy5-labeled Eap1 protein (final concentration of in the vaccines at 1 mg/mL) with the same procedure as described above. The three kinds of vaccines, including AlgelCpG+Cy5-Eap1, AlgelCpG+Cy5-Em, AlgelCpG+Cy5-EmEAP1, were subcutaneously injected into the hind footpad of the mice at the dose of 100 μL vaccine per mouse. At the different time points after injection, the mice were anesthetized by using ether. The distribution of the Cy5 fluorescence in the mice was detected by a small animal fluorescence imaging system (Xenogen, USA).

2.5. Lymph node weighing and dendritic cell assays

To investigate the effect of the vaccine on lymph node growth and dendritic cell accumulation, the mice were immunized by the three kinds of vaccines (100 μL vaccine per mouse) with the prime-boost procedure on day 0 and day 7. The mice were euthanized by CO2 on day 14. Both the inguinal and cervical lymph nodes were sampled from the mice and weighed. The lymph node tissues were grinded, and the total lymph node cells were obtained by filtering with 100-mesh sieves to remove cell aggregates. The cells were then stained by the FITC-conjugated CD11c antibody. The percent of CD11c-positive dendritic cells were measured by using flow cytometry (BD, USA).”

6 Splenocyte assays 138 The vaccinated mice were euthanized on day 14 as described above, and the 139 spleens of the mice were sampled. The splenocytes were prepared also by grinding and 140 filtering. The obtained splenocytes were stained by the FITC-conjugated CD4 antibody 141 and the APC-conjugated anti-mouse IL-2 and IFN-γ antibodies. The cells were then ex- 142 amined by using flow cytometry [30]. 143 144 2.7. ELISA assays 145 For determination of the serum anti-Eap1 IgG levels, the venous blood was sampled 146 from the tail veins of the vaccinated mice on day 14, day 28 and day 56. After storage at 4 147 °C for 12 h, the blood was centrifuged at 1500 rpm for 5 min, obtaining the serum for 148 ELISA assays. The serum was diluted by PBS at a two-fold gradient, and the diluted so- 149 lutions were( added into the 96-well plates pre-incubated with Eap1 solution (100 μg/mL). 150 The bound IgG in the wells was further incubated by the HRP-conjugated anti-mouse IgG 1(its not clear please modify the above information)

Response:

These method section has been revised (Page 3, Lines 131-147):

2.5. Lymph node weighing and dendritic cell assays

To investigate the effect of the vaccine on lymph node growth and dendritic cell accumulation, the mice were immunized by the three kinds of vaccines (100 μL vaccine per mouse) with the prime-boost procedure on day 0 and day 7. The mice were euthanized by CO2 on day 14 [32]. Both the inguinal and cervical lymph nodes were sampled from the mice and weighed. The lymph node tissues were grinded, and the total lymph node cells were obtained by filtering with 100-mesh sieves to remove cell aggregates. The cells were then stained by the FITC-conjugated CD11c antibody. The percent of CD11c-positive dendritic cells were measured by using flow cytometry (BD, USA).

2.6. Splenocyte assays

  The vaccinated mice were euthanized on day 14 as described above, and the spleens of the mice were sampled. The splenocytes were prepared also by grinding of the spleen tissues and filtering to remove aggregates. The obtained splenocytes were stained by the FITC-conjugated anti-CD4 antibody, together with the APC-conjugated anti-mouse IL-2 antibody or the APC-conjugated anti-mouse IFN-γ antibody. The cells were then examined by using flow cytometry [30].”

7 please check plagiarism)

Response:

The manuscript has been carefully checked for avoiding plagiarism.

8 please mention methodology clearly

Response:

The Method section has been revised for clear expression.

9 Please check plagiarism

Response:

The manuscript has been carefully checked for avoiding plagiarism.

10 Please arrange references as per medeley and Zotero

Response:

The references have been carefully checked according to the author introduction of the journal.

11 please follow author guidelines

Response:

The manuscript has been carefully checked according to the author introduction of the journal.

Thank you very much again for your great help to this manuscript.

Reviewer 2 Report

The authors present a new vaccine candidate against C. albicans. This is a high interest field since fungal infections are of major concern specially among immunocompromised patients. The work is overall very well planned and presented but I would like to make some comments to the authors:

1) In the statistical analysis you mention three replicates (n=3). Do you mean you repeated the experiment three times? If so, for the mice work you used 13 x 3 mice/group?

2) Figure 1 and 2. What are FLU units? Fluorescent? Check if there is a clearer way to express this units.

3) Line 215. Please clarify how in Fig 2c, lower fluorescent signal means higher vaccine stability. It was confusing to understand since you'd like to have higher signal to prove vaccine stability over time. I figure it is because it is in the supernatant so it was released right? Please,  add a sentence that clarifies this result.

4) Why did you only compared kidneys? I would suggests to add other organs such as blood, liver, spleen (CFU, weight) and brain.

5) I'd suggest to mention some limitations of this study. For example, the time points seem very short for me, specially talking about immunity, the ultimate goal is to provide long term protection. Also, the CFU reductions despite being statistically significant it is only 2 units below control... which in practice, the patient would still be sick. Please discuss the study limitations and future work.

6) In the discussion, it would be appreciated a broader discussion comparing other fungal vaccine candidates in both Candida sp. or other species. 

Author Response

Response to Reviewer 2

The authors present a new vaccine candidate against C. albicans. This is a high interest field since fungal infections are of major concern specially among immunocompromised patients. The work is overall very well planned and presented but I would like to make some comments to the authors:

1) In the statistical analysis you mention three replicates (n=3). Do you mean you repeated the experiment three times? If so, for the mice work you used 13 x 3 mice/group?

Response:

We repeated most of the experiments three times, including Splenocyte assays, ELISA assays, anti-Eap1 IgG levels, the number of fungi accumulated in the kidneys, and TNF-α level assay, etc. However, the mouse systemic infection was performed with 13 replicates. The description has been revised (Page 4, Lines 172-173):

“Most of the experiments were performed with three replicates (n = 3), expect the experiment of mouse vaccination and systemic infection (n = 13).”

2) Figure 1 and 2. What are FLU units? Fluorescent? Check if there is a clearer way to express this units.

Response:

Thanks for your insightful suggestion. The FLU unit indicates the fluorescence intensity of the samples, which is detected by a fluorescence microplate reader to reflect whether synthetic proteins are expressed. The details have been supplied in the Method section and the figure legend (Page 3, Lines 105-106; Page 6, Lines 235-236):

“The fluorescence intensity units (FLU) of FITC of the stained cells was determined by using a fluorescence microplate reader (Enspire, PerkinElmer, USA).”

Figure 2. Preparation protocol and characterization of the AlgelCpG+Em and AlgelCpG+EmEAP1 vaccines. (a) A scheme illustrating the preparation of the gel vaccines. (b) Confocal images of the gels after 28 days of storage. (c) Fluorescence intensity of the released supernatants by centrifugation of the gels after storing for indicated time. The FLU values of the supernatant were detected by a fluorescent microplate reader. The asterisks indicate significant difference between the two groups at each time point (P < 0.05).”

3) Line 215. Please clarify how in Fig 2c, lower fluorescent signal means higher vaccine stability. It was confusing to understand since you'd like to have higher signal to prove vaccine stability over time. I figure it is because it is in the supernatant so it was released right? Please, add a sentence that clarifies this result.

Response:

Thank. We have added a sentence to clarify the result (Page 5, Lines 222-223):

“For instance, after 112 days of storage, AlgelCpG+Em released 1.79-fold lower levels of the fluorescence protein than AlgelCpG+EmEAP1 (Figure 2c), indicating that AlgelCpG+EmEAP1 had high stability and maintained the persistence of mCherry fluorescent protein and Eap 1 in synthetic cells.”

4) Why did you only compared kidneys? I would suggests to add other organs such as blood, liver, spleen (CFU, weight) and brain.

Response:

Thank you very much for your positive comments and constructive suggestions to our manuscript. The kidney is the main targeted organ of C. albicans systemic infections. We supplied the background and a representative reference (Page 9, Lines 333-334; Page 14, Line 565):

“The kidney is the primary site of systemic infection by C. albicans, and its fungal burden could reflect the degree of systemic infection by this pathogen [47,48].”

“48. Lopes, J.P.; Lionakis. M.S. Pathogenesis and virulence of Candida albicans, Virulence 2023, 13, 89-121.”

5) I'd suggest to mention some limitations of this study. For example, the time points seem very short for me, specially talking about immunity, the ultimate goal is to provide long term protection. Also, the CFU reductions despite being statistically significant it is only 2 units below control... which in practice, the patient would still be sick. Please discuss the study limitations and future work.

Response:

Thank you for your valuable comments and important questions. We discussed the limitations of this study in the manuscript (Page11, Lines 424-431):

“In this study, we showed that AlgelCpG+EmEAP1 remained stable and resided in the lymph nodes of mice in 112 days after injection. However, the long-term durability of the vaccine and its efficiency in protecting patients from recurrent and multi-drug resistant Candida albicans remain to be investigated. In the meantime, it is suggested to further improve the capacity of the vaccine to reduce the fungal burden for reduction of the infectious chances in patients. In recent years, the development of new antifungal therapeutics remains slow [17]. Therefore, our further investigations will focus on promotion the efficiency of the fungal vaccines for real clinical application.”

6) In the discussion, it would be appreciated a broader discussion comparing other fungal vaccine candidates in both Candida sp. or other species. 

Response:

Thank you for your valuable comments. A broader discussion comparing the vaccine candidates has been supplied (Page 10, Lines 376-381; Page 14, Linex 574-578):

“In recent years, fungal extracellular vesicles carrying immunogenic proteins, such as mannan protein MP88, chitin deacetylase Cda family proteins, and Vep proteins, have been used for vaccine preparation against fungal pathogens such as Candida and Paracoccidioides [53,54]. However, due to the fact that fungal extracellular vesicles are a mixture of multiple components, some of which may have toxic effects, there are obstacles to develop vesicle-based vaccines for application [55].”

“Loh, J.T.; Lam, K.P. Fungal infections: Immune defense, immunotherapies and vaccines. Adv. Drug Deliver. Rev. 2023, 196, 114775.

Rizzo, J.; Chaze, T; Miranda, K; Roberson, R.W.; Gorgette, O.; Nimrichter. L.; Matondo. M.; Latgé, J.; Beauvais, A.; Rodrigues, M.L. Characterization of extracellular vesicles produced by Aspergillus fumigatus protoplasts. mSphere 2020, 5, 10-1128.

Rizzo, J.; Rodrigues, M.L.; Janbon, G. Extracellular vesicles in fungi: past, present, and future perspectives. Front. Cell. Infect. Microbiol. 2020, 10-346.”

Thank you very much again for your great help to the manuscript.

Reviewer 3 Report

Summary

Researchers used synthetic biology techniques to create a synthetic cell called EmEAP1, which displayed a specially designed protein called mChEap1 on its surface. This synthetic cell, when induced with IPTG, effectively presented the artificial protein. The fungal vaccine was then formulated by combining EmEAP1 cells with aluminum hydroxide gel and CpG. The vaccine remained stable even after 112 days of storage. When administered to mice, the vaccine accumulated in lymph nodes and triggered the recruitment of CD11c+ dendritic cells. Additionally, it strongly activated CD4+ T splenocytes and generated high levels of anti-Eap1 IgG antibodies. Through prime-boost immunization, this vaccine extended the survival time of mice infected with Candida albicans cells and reduced fungal colonization and kidney inflammation.

Methods

Only two references in methods? This should be increased

Authors could use material from these papers for improving introduction/discussion

https://www.frontiersin.org/articles/10.3389/fmicb.2012.00294/full

https://www.ncbi.nlm.nih.gov/pmc/articles/PMC7558412/

Caption of figure 3 is confusing. Authors state imaging of the Cy 5-labeled antigen Eap1 (a) and alteration of lymph node indicators (b,c) in the mice receiving vaccination. We cannot see labels with mouse diagrams, maybe due to black background

Discussion should elaborate on kidney inflammation reduction due to this vaccine as well.

Language should be edited for mistakes.

Minor english check is required.

Author Response

Response to Reviewer 3

Researchers used synthetic biology techniques to create a synthetic cell called EmEAP1, which displayed a specially designed protein called mChEap1 on its surface. This synthetic cell, when induced with IPTG, effectively presented the artificial protein. The fungal vaccine was then formulated by combining EmEAP1 cells with aluminum hydroxide gel and CpG. The vaccine remained stable even after 112 days of storage. When administered to mice, the vaccine accumulated in lymph nodes and triggered the recruitment of CD11c+ dendritic cells. Additionally, it strongly activated CD4+ T splenocytes and generated high levels of anti-Eap1 IgG antibodies. Through prime-boost immunization, this vaccine extended the survival time of mice infected with Candida albicans cells and reduced fungal colonization and kidney inflammation.

Response:

Thank you very much for your positive comments and constructive suggestions to our manuscript. We have carefully revised the manuscript according to your suggestions, and respond to them point by point.

Methods

1) Only two references in methods? This should be increased.

Response:

Some additional references have been supplied to support the methods (Page 3, Line 135; Page 4, 153; Page 13, Lines 530-531, Lines 534-536):

“The mice were euthanized by CO2 on day 14 [32].”

“After stored at 4 °C for 12 h, the blood was centrifuged at 1, 500 rpm for 5 min, obtaining the serum for ELISA assays [34].”

“32. Zhang, Q.; Hub, M.; Xu, L.; Yang, X.; Chang, Y.; Zhu, Y. Effect of edible fungal polysaccharides on improving influenza vaccine protection in mice. Food Agr. Immunol. 2017, 6, 981–992.”

“Yam, K.; Gupta, J.; Winter, K.; Allen, E.; Brewer, A.; Beaulieu, E.; Mallett, C.; Burt, D.; Ward, B. AS03-adjuvanted, very-low-dose influenza vaccines induce distinctive immune responses compared to unadjuvanted high-dose vaccines in BALB/c mice. Front. Immunol. 2015, 6, 207.”

2) Authors could use material from these papers for improving introduction/discussion

https://www.frontiersin.org/articles/10.3389/fmicb.2012.00294/full

https://www.ncbi.nlm.nih.gov/pmc/articles/PMC7558412/

Response:

The above papers have been cited in the introduction (Page 2, Lines 47-48; Page 12, Lines 492-494):

“Nowadays, there are a few kinds of preventive fungal vaccines that have been developed [13-16].”

“13. a, L.; Taborda, C.; Nosanchuk, J. Advances in Fungal Peptide Vaccines. J. Fungi 2020, 6, 119.”

“14. Vecchiarelli, A.; Pericolini, E.; Gabrielli, E.; Pietrella, D. New approaches in the development of a vaccine for mucosal candidiasis: progress and challenges. Front. Microbiol. 2012, 3, 294.”

  • Caption of figure 3 is confusing. Authors state imaging of the Cy 5-labeled antigen Eap1 (a) and alteration of lymph node indicators (b,c) in the mice receiving vaccination. We cannot see labels with mouse diagrams, maybe due to black background

Response:

Thank you for your valuable comments. We revised the caption for clarifying the figures (Page 7, Lines 260-261):

Figure 3. Imaging of the Cy 5-labeled antigen Eap1 (a), alteration of lymph node weights (b), and change of CD11c+ cells (c) in the mice receiving vaccination.”

  • Discussion should elaborate on kidney inflammation reduction due to this vaccine as well.

Response:

Thanks for your insightful suggestion. The discussion for reduced kidney inflammation by the vaccine has been supplied (Page 11, Lines 406-415):

“By measuring kidney fungal burden, kidney TNF-α levels and kidney weights on day 3, we found that the fungal burden was proportional to the TNF-α levels and kidney weights, that is, the higher fungal numbers, the higher inflammatory factor levels and kidney weights. This indicates that the vaccine could attenuate fungal infection and colonization in the kidneys, thereby alleviating the inflammatory response in the kidneys (Figure 5b,c,d). This may be because the AlgelCpG+Cy5-EmEAP1 vaccine evoked high levels of the fungus-specific IgG, leading to strong humoral immunity against the fungal cells invading the kidneys. Consequently, the mice receiving the vaccination of AlgelCpG+Cy5-EmEAP1 exhibited much weaker kidney inflammation than the control mice which suffered from severe fungal infections in kidneys.

” 

5) Language should be edited for mistakes.

Response:

Thank you for your valuable comments. We have made careful language revisions throughout the manuscript.

Thank you very much again for your great help to the manuscript.